# Histopathological Patterns of Cutaneous Adverse Reaction to Anti-SARS-CoV-2 Vaccines: The Integrative Role of Skin Biopsy

**DOI:** 10.3390/vaccines11020397

**Published:** 2023-02-09

**Authors:** Gerardo Cazzato, Francesca Ambrogio, Maria Carla Pisani, Anna Colagrande, Francesca Arezzo, Eliano Cascardi, Miriam Dellino, Enrica Macorano, Irma Trilli, Paola Parente, Teresa Lettini, Paolo Romita, Andrea Marzullo, Giuseppe Ingravallo, Caterina Foti

**Affiliations:** 1Section of Molecular Pathology, Department of Precision and Regenerative Medicine and Ionian Area (DiMePRe-J), University of Bari “Aldo Moro”, 70124 Bari, Italy; 2Section of Dermatology and Venereology, Department of Precision and Regenerative Medicine and Ionian Area (DiMePRe-J), University of Bari “Aldo Moro”, 70124 Bari, Italy; 3Section of Gynecology and Obstetrics, IRCSS Giovanni Paolo II, 70124 Bari, Italy; 4Department of Medical Sciences, University of Turin, 10124 Turin, Italy; 5Pathology Unit, FPO-IRCCS Candiolo Cancer Institute, Str. Provinciale 142 lm 3.95, 10060 Candiolo, Italy; 6Section of Legal Medicine, Department of Interdisciplinary Medicine, University of Bari “Aldo Moro”, 70124 Bari, Italy; 7Odontomatostologic Clinic, Department of Innovative Technologies in Medicine and Dentistry, University of Chieti “G. D’Annunzio”, 66100 Chieti, Italy; 8Pathology Unit, Fondazione IRCCS Ospedale Casa Sollievo della Sofferenza, 71013 San Giovanni Rotondo, Italy

**Keywords:** vaccines, cutaneous adverse reactions, ADR, SARS-CoV-2, COVID-19, dermatopathology, dermatology

## Abstract

The advent of vaccines represented a milestone to allow the slowing down and then containing of the exponential increase in ongoing infections and deaths of COVID-19. Since the first months of the vaccination campaign in various continents, there has been a certain number of reports of adverse events, including skin reactions. We conducted a systematic review, searching on PubMed, Web of Science, Scopus, and Cochrane Library for the words: COVID vaccine, dermatopathology, skin, eruptions, rash, cutaneous, BNT162b2 (Pfizer-BioNTech), ChAdOX1 (AstraZeneca), and mRNA-1273 (Moderna). A total of 28 records were initially identified in the literature search of which two were duplicates. After screening for eligibility and inclusion criteria, 18 publications were ultimately included. Various clinical cutaneous manifestations and histopathological patterns following vaccination have been described in literature. The most frequent clinical-pathological presentations were erythematous maculo-papular eruptions in different way of distribution with histopathological pictures mostly represented by interface changes and mixed peri-vascular and peri-adnexal cell infiltrate. Other presentations included new onset of pemphigoid bullous disease (n = 15), delayed T-cell-mediated hypersensitivity reaction (injection site reactions) (n = 10), purpuric skin rash (n = 13), mostly localized on the legs bilaterally and symmetrically with histological pictures characterized by extravasation of erythrocytes in the superficial and middle dermis, and other types of reactions. New studies with large case series and further literature reviews are needed to improve the clinical management of patients and optimize the timeline for carrying out histological biopsy for confirmatory, supportive, and differential diagnosis purposes.

## 1. Introduction

For almost three years now, the SARS-CoV-2 pandemic has put a strain on the health systems of the entire planet, with human, social, economic, and environmental damages that have changed the pre-pandemic balance [1]. From the earliest months of the pandemic, it was understood that COVID-19 was not only a disease of respiratory relevance (lung and respiratory tree) but also had the potential to affect a multitude of other body districts, including the cardiovascular [2], reproductive [3], as well as integumentary systems. The advent of vaccines represented, on the other hand, a milestone to allow the slowing down (at first) and then containing of the exponential increase in ongoing infections and deaths of COVID-19 [2]. Globally, as of 4:23 pm CET, 2 December 2022, there have been 640,395,651 confirmed cases of COVID-19, including 6,618,579 deaths, reported to World Health Organization (WHO). As of 30 November 2022, a total of 13,042,112,489 vaccine doses have been administered [3]. Since the first months of the vaccination campaign in various continents, there has been a certain number of reports of adverse events, including skin reactions [4]. The most frequent adverse effects that were reported concerned pain, swelling, or redness at the site of injection, fever, fatigue, headache, muscle pain, nausea, vomiting, chills, itching, joint swelling and/or pain, and rarely anaphylactic shock [4,5,6]. Among skin reactions, the most frequently reported were injection site reactions (30–70% of cases) and generally of mild/moderate intensity, but there were also some case reports and/or case series which included filling reactions, exanthemas, vascular lesions, urticaria, eczematous dermatitis, bullous autoimmune dermatitis onset, and severe cutaneous adverse reactions. Furthermore, during the months of intense vaccination activity, the exacerbation of chronic immune-mediated dermatoses (mainly represented by atopic dermatitis, AD, and psoriasis) and possible reactivations of the viruses, such as herpes infection (HSV), have been reported [7,8]. The skin reactions were generally mild, self-limiting, and resembled common drug rashes and/or COVID-19 skin manifestations [9,10]. Although many reactions from a clinical point of view have been reported in the literature, there has been much less information and reports regarding the histological pattern underlying these eruptions. In this review, we focus exclusively on case reports, case series, research articles, and reviews that have reported one or more cutaneous manifestations with a histological counterpart; we, therefore, try to provide a picture of the state of the art, and we try to outline possible future perspectives in this regard.

## 2. Materials and Methods

To investigate the various histopathological features of the cutaneous adverse reaction and to try to correlate to the potential mechanisms underlying SARS-CoV-2 vaccine-related dermatological manifestations, a systematic review was performed according to the Preferred Reporting Items for Systematic Reviews and Meta-Analyses (PRISMA) guidelines [11]. Searches on PubMed, Web of Science (WoS), Scopus, and Cochrane Library were conducted using the combination of the following keywords and medical subject heading (MeSH) terms: COVID vaccine, dermatopathology, skin, eruptions, rash, cutaneous, BNT162b2 (Pfizer-BioNTech), ChAdOX1 (AstraZeneca), and mRNA-1273 (Moderna). The analyzed period of our search was from 1 March 2020 to 3 December 2022. We included only articles in the English language, and all types of papers were analyzed, such as case reports, letters to the editor, case series, multicentric studies, reviews, and registry-based studies.

The inclusion criteria required that the manuscripts contain information on the type of vaccine administered (Pfizer-BioNTech, Moderna, J&J/Janssen, Covaxin, AstraZeneca, and CoronaVac vaccines, both first and second doses), and at least one biopsy was performed for histopathological confirmation. Two authors (G.C. and F.A.) independently assessed the risk of bias of each included study in accordance with methods recommended by the National Institutes of Health Quality Assessment Tool for Case Series Studies [12]. Authors resolved disagreements by a discussion, and a third author (G.I.) was consulted to resolve disagreements when the situation needed. The full texts of analyzed studies were reviewed, and data on authors, article type, number of presented cases with cutaneous involvement, eruptions type (clinical/diagnosis information), and histopathological features were extracted from articles texts, tables, figures, and are summarized in Table 1.

## 3. Results

A total of twenty-eight records were initially identified in the literature search of which two were duplicates. After screening for eligibility and inclusion criteria, 19 publications were ultimately included (Figure 1).

The majority of publications were letters to editors (n = 6) and case reports (n = 6) followed by case series (n = 3), observational multicentric studies (n = 2), and an original article (n = 1). A total of 131 patients with dermatologic symptoms and almost one biopsy with histopathological investigations were included. The features of the patients of the studies included in this review were reported in Table 1.

Eighty-two patients who experienced at least one adverse reaction and underwent at least one confirmatory skin biopsy were reported. Among them, eighteen patients reported the onset of an autoimmune disorder, represented in most cases (n = 16) by new onset of disease of the pemphigoid/pemphigus bullous family, one case of onset of psoriasis guttata not previously reported in the patient’s medical history, and one case of Rowell syndrome, a very rare form of lupus erythematosus. The highest rate of adverse skin reactions occurred after administration of mRNA vaccines compared with adenovirus vector vaccines.

Various clinical cutaneous manifestations and histopathological pattern following vaccination have been described in the literature. The most frequent clinical–pathological presentations were erythematous maculo-papular eruptions with different ways of distribution, with or without itchiness (n = 20), near or at a distance to the injection site, with histopathological pictures mostly represented by interface changes, and mixed perivascular and peri-adnexal cell infiltrate. The secon most frequent presentation was the new onset of pemphigoid bullous disease (n = 16) with subepidermal/intraepidermal blisters with eosinophils. Thirdly, there were fifteen cases of vaccine-related eruptions of papules and plaques (V-REPP) with spongiotic phenomena and different grades of intensity of interface change (n = 15); delayed T-cell-mediated hypersensitivity reactions (injection site reactions) (n = 10) characterized by mixed-cell infiltrate/spongiotic dermatitis with eosinophils and also some mast cells; purpuric skin rash (n = 13), mostly localized on the legs bilaterally and symmetrically with histological pictures characterized by extravasation of erythrocytes in the superficial and middle dermis; nine cases described a picture of eczematous dermatitis (n = 9), histologically with spongiotic pattern; urticarial eruptions (or vasculitis) were described in seven cases (n = 7). Furthermore, pernio-like lesions (n = 5), erythema multiforme-like lesions (n = 5), vasculitis (n = 5), chilblain-like eruptions (n = 3), neutrophilic dermatosis (n = 2), and also more rare manifestations, such as Steven–Johnson syndrome (SJS), fixed drug eruptions, and possible release of previous autoimmune conditions, such as Rowell’s syndrome (very rare type of erythematous lupus), were described.

## 4. Discussion

In this paper, we conducted a systematic literature review analyzing only cases of vaccine adverse reactions (both mRNA and vector-mediated of adenovirus) in which at least one diagnostic biopsy was performed. The major adverse reaction in this cohort of patients was represented by maculo-papular eruptions hours/days following the administration of the first or second dose of vaccine at the injection site or at a distance. This type of rash overlaps, in some aspects, with the adverse reactions from vaccines at the injection site only (which came in third place in our systematic review) and can be explained both in terms of delayed type T-cell-mediated hypersensitivity reaction and as a potential reaction to vaccine adjuvants, such as polyethylene glycol-2000 (PEG) in the Pfizer-BioNTech vaccine, polyethylene glycol-2000 and tromethamine in the Moderna vaccine, and polysorbate 80 in the AstraZeneca vaccine [32]. In a comprehensive review by Gambichler et al. [31], the issue of the two types of hypersensitivity reaction of immediate type (or type I) and delayed type (or type II) is addressed in detail. From the analysis of published cases up to the time of the writing of their review, the authors inferred how type I reactions, including urticaria, angioedema, and even anaphylactic reactions, by virtue of their importance and potential dangerousness, constitute a reason for changing the vaccination plan with non-mRNA vaccines, given and considering that this type of vaccine has been shown to be a greater harbinger of such adverse reactions [33]. Relative to type IV reactions, the major symptoms/signs reported in the literature have been erythema, induration, and pain at the injection site, with a timeline averaging about one week, with no particular risks contraindicating any future booster administrations [34].

It is of great interest to find that a number of patients presented “de novo” onset of bullous dermatosis of the bullous pemphigoid (PB) type after vaccination with an anti-SARS-CoV-2 vaccine. This reaction, although not yet unequivocally established as related to the vaccine, has been reported in various works. The two major theories accredited to explain these manifestations are represented by a mechanism of “molecular mimicry”, which would mediate a “cross-link” between the peptide structures of the SARS-CoV-2 spike protein and the host’s own antigenic structures [35] or from a genetically based predisposition relating to immune/autoimmune dysregulation, which, in the presence of nucleic acids introduced with the vaccine, would lead to a reaction of this type.

Regarding chilblain-like reactions after anti-SARS-CoV-2 vaccine, it is important to remember how the earliest descriptions of skin reactions during COVID-19 infectionwere related precisely to acral-distributed chilblain lesions [36]. This type of lesion has been investigated particularly in an attempt to understand whether it is really due to direct penetration of the virus or parts of the spike glycoprotein into the skin (excretory portion of the ducts of the eccrine sweat glands, vascular endothelium of the superficial dermal papillary plexus, etc.) rather than related to immune-based reactions related to likely dysregulation during COVID-19 infection [37]. No unambiguous answer has yet been arrived at due to the presence of discordant results, but in some reports of adverse reactions to vaccine, reference has been made to this type of adverse event.

Another important question raised concerns the possible reactivation of previous infections (such as herpes virus, varicella zoster virus, etc.) following administration of anti-SARS-CoV-2 vaccines. As previously stated by Gambichler et al. [31], there are no certainties in this regard in that while one can speculate that viral particles are capable of inducing reactivation of certain types of viruses, causing a pityriasis rosea (PR) and PR-like skin reaction pattern, the high prevalence of such infectious diseases and multiple predisposing conditions (immunosuppressive therapies, malignant neoplasms) could distort the epidemiological link underlying such reports [16].

In the field of purpuric/vasculitic manifestations following vaccine administration, there have been various attempts to provide a convincing etiopathogenesis that can lay a valid basis for explanation. In this regard, reports of such reactions have included the possibility of the occurrence of a vaccine-related thrombocytopenia condition due to potential factors, including immunological dysregulation, molecular mimicry, cryptic antigen expression on platelets, and so on [37]. Fujita et al. proposed a potential explanation behind these purpuric-type reactions in patients undergoing vaccination, in that considering the epidemiological data, certain impurities contained in the coding RNA may elicit a mechanism of immune dysregulation with an autoimmune background going on to orchestrate an immunologic response against platelet factors and subsequent hemorrhagic/purpuric manifestations [38].

In the field of very rare adverse reactions, some authors have reported anecdotal cases of Steven–Johnson syndrome (SJS) in patients who had received the first administration of ChAdOx1 nCoV-19 (AstraZeneca). Specifically, Dash et al. [20] described a case of a 60-year-old patient, polymedicated for hypertension with amlodipine and for diabetes with metformin and teneligliptin, who within three days following vaccine administration, had developed a widespread skin rash with the presence of a few blisters over the entire body and whose biopsy reported the presence of histopathologic features consistent with SJS. In an attempt to correlate this manifestation with the vaccine, the authors made use of an algorithm (Naranjo’s algorithm) from which an association of the value of 2 resulted, i.e., as a possible adverse reaction to the vaccine. In an attempt to explain the mechanisms of etiopathogenesis, the authors reiterated the theses of Chahal et al. [39] in which they cited the possibility that virotopes expressed on the surface of keratinocytes were capable of eliciting a CD8+ T lymphocyte response specific to epidermal cells, resulting in apoptosis of keratinocytes and detachment of the dermo-epidermal junction. Furthermore, it seems the AstraZeneca vaccine enhances the lymphocyte Th1 CD4+ immunological response, leading to the possibility to cause this kind of cutaneous eruption.

The same author, in another paper, first described a case of Rowell syndrome in a 74-year-old woman who presented erythematous partly violaceous coalescing macules and papules with slightly indicated cocarde formation on the trunk and extremities. Skin biopsies taken showed epidermal atrophy and a vacuolar interface dermatitis with lymphocytic infiltrates along the dermo-epidermal junction associated with dyskeratoses of basal keratinocyte. Serology revealed antinuclear autoantibodies (ANA) with 1:640 in speckled pattern as well as positivity for anti-Ro/SSA, anti-Ro/SSA, and anti-La/SSB antibodies. The clinical and pathologic features were suggestive of this type of LE, and the lack of other comorbidities and triggers supported the possible association between BNT162b2 vaccine administration and onset of this eruption [40].

From what has already been said by other colleagues [7], the adverse reactions from anti SARS-CoV-2 vaccines (both mRNA and adenovirus vector) recapitulate some of the ongoing cutaneous manifestations of COVID-19 [7,41,42].

From the analysis of these works, it is quite clear the most common reactions (local and delayed skin reactions, such as urticaria, maculopapular, or non-specific skin rashes) have already been previously described with the use of other vaccines, such as the vaccine against hepatitis B [34].

In any case, most of these adverse skin reactions were self-limiting and resolved quickly, except for some rarer manifestations, such as Stevens–Johnson syndrome in which the speed and extent of manifestations can pose a risk itself to life. In these cases (results of only two from our review) it is essential to ask whether it is appropriate to proceed with a booster dose or postpone the administration of any second/third dose of the vaccine [43].

In all the studies analyzed, the skin biopsy proved to be an important element for an accurate differential diagnosis and as a support tool for the subsequent clinical decision. In addition, it is worth noting that by allowing discrete amounts of skin tissue to be harvested, histological biopsy is an absolutely important means of performing further scientific research investigations aimed at improving the understanding of certain skin phenomena or epiphenomena.

Finally, it is very important to emphasize how much the performance of a confirmatory skin biopsy must follow detailed rules in terms of procedure, site of performance, and presence of comprehensive clinical information. In fact, as with other areas of inflammatory dermatopathology, a skin biopsy (whether performed by excision, incision, or shave/punch) should always be performed in an area affected by the manifestation/pathology one wishes to investigate further and when possible, consider also performing a biopsy on a portion of healthy skin for possible comparisons.

In addition, it is a good idea to always accompany the card with a detailed anatomic-clinical history as even potential medications implemented could mask or alter potentially vaccine-related manifestations.

### Limitations

The major limitation of our systematic review is that in an attempt to offer the reader a “histopathological” view of the problem of adverse skin reactions after administration of anti-SARS-CoV-2 vaccine, a very small number of patients were examined. In addition, the prevalence rates of the various skin manifestations may have deviated from the most frequently reported values for this study design and therefore, should always be supplemented and considered in light of how the review was conducted. Finally, the inclusion of English language articles among the inclusion criteria may have invalidated the possibility of retrieving other reports written in other languages, mainly (but not only) from China.

## 5. Conclusions

New studies with large case series and further literature reviews are needed to better understand which are the most common or the rarest adverse skin manifestations from SARS-CoV-2 vaccines, improve the clinical management of patients, and optimize the pathway for carrying out histological biopsy for confirmatory, supportive, and differential diagnosis purposes. The role of skin biopsy during anti-SARS-CoV-2 vaccine adverse reactions continues to be of importance not only for clinical/diagnostic purposes but also for applied research in an attempt to elaborate information related to the major patterns described as well as their relationship to the skin manifestations during COVID-19 infection, as previously described.

## Figures and Tables

**Figure 1 vaccines-11-00397-f001:**
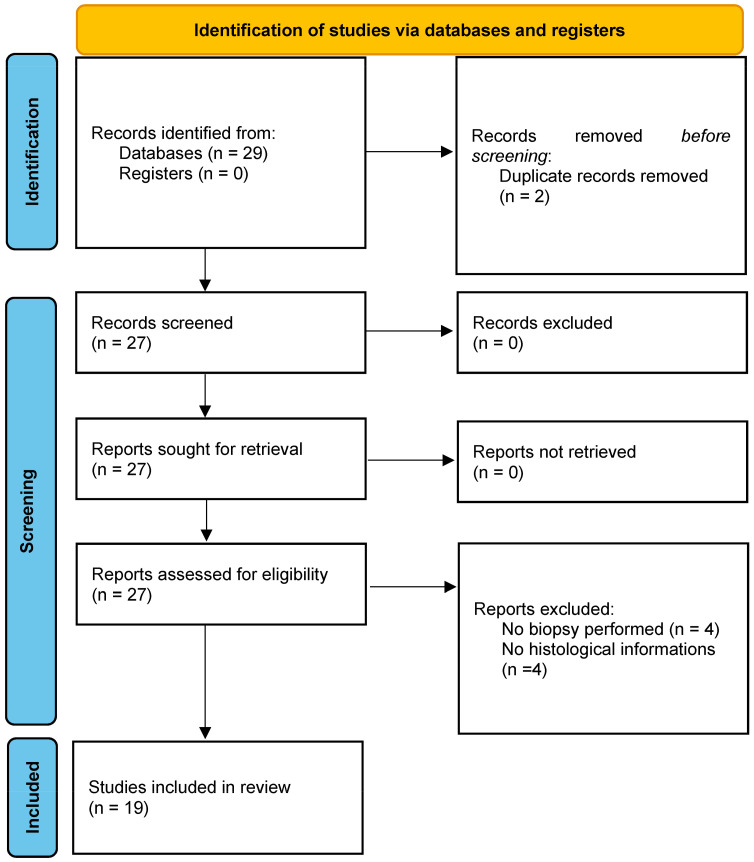
Articles selection and flow-chart of PRISMA guidelines used in this review.

**Table 1 vaccines-11-00397-t001:** Demographical, clinical, histopathological, and diagnostic features of some studies included in this review.

Author(s)	Type of Article	Number of Patients	Histopathological Features	Clinical Presentation/Diagnosis
Larson et al. [13]	Original Article	12 (6 F-6 M)	2 cases: mixed-cell infiltrate/spongiotic dermatitis with eosinophils 3 cases: interface changes as exanthematous drug reaction 3 cases: spongiotic pattern 2 cases: small-vessels vascular injury 2 cases: subepidermal/intraepidermal bulla and eosinophils	injection-site reactions generalized eruptions eczematous dermatitis 1 case: urticarial vasculitis 1 case: leukocytoclastic vasculitis new-onset bullous pemphigoid
Papadimitrious et al. [14]	Letter to editor	2 (with biopsy)	moderate perivascular lymphocytic infiltrate, mild dermal focal oedema with red cell extravasation, and rare interstitial eosinophils	delayed localized hypersensitivity reaction
Annabi et al. [15]	Case series	8 (3 M-5 F)	2 cases: spongiotic dermatitis 1 case: dermal edema, perivascular and peri-eccrine lymphocytic infiltrate 1 case: vacuolar interface dermatitis 1 case: epidermal dysmaturation, vacuolization of basal keratinocytes, apoptotic cells 1 case: vacuolar interface dermatitis, perivascular superficial lymphocytic infiltrate with numerous eosinophils 1 case: lichenoid interface dermatitis 1 case: dermal edema and perivascular lymphocytic infiltrate	morbilliform rash erythematous nodules/chilblains diffuse erythematous rash livedo racemosa of thighs fixed drug eruption diffuse maculopapular pustular exanthema edematous plaque of the buttock and thigh
Avallone et al. [16]	Multicentric study	9 (5 M-4 F) with biopsy	1 case: spongiotic and interface vacuolization 1 case: interface change and a dense band-like lichenoid lymphocytic infiltrate 2 cases: vacuolar damage, extravasated erythrocytes 1 case: apoptotic keratinocytes, vacuolar damage and perivascular lymphocytic infiltrate 1 case: spongiotic pattern, exocytosis, perivascular lymphocytic infiltrate 1 case: psoriasiform-like pattern1 case: subepidermal blisters 1 case: small-vessels flogistic infiltration	maculo-papular rash lichenoid drug eruption chilblain-like erythema multiforme-like pytitiasis rosea-like eruptions eczematous drug eruptions bullous pemphigoid cutaneous lymphocytic vasculitis
Magro et al. [17]	Case series	22 (10 F-12 M)	9 cases: lymphocytes-mediated interface dermatitis 6 cases: eczematous change with some eosinophils 2 cases: vasculitic presentation 3 cases: acanthosis with spongiosis 1 case: acantholytic diskeratosis 1 case: psoriasiform-pattern	generalized papulo-vescicular eruptions eczematous dermatitis pernio-like erythema urticarial eruptions Grover disease guttate psoriasis
McMahon et al. [18]	Registry-based study	58 (36 F-22 M)	15 cases: **3** robust V-REPP > marked spongiosis with intraepidermal vescicles and minimal to no interface changes **8** moderate V-REPP > moderate spongiosis more often than interface change **4** mild V-REPP > mild spongiosis and more prominent interface changes	V-REPP
			12 cases: subepidermal/intraepidermal vescicles with eosinophils 4 cases: perivascular lymphocytic infiltrates with eosinophils and mast cells 4 cases: viral cytopathic changes4 cases: interface dermatitis 3 cases: vasculitic presentation 2 cases: acanthosis with spongiosis 2 cases: neutrophilic infiltrate 2 cases: small-vessels injury with nuclear dust 2 cases: perivascular lymphocytic inflammation 2 cases: perivascular lymphocytic inflammation 1 case: erythromelalgia 5 others (Steven Johnson syndrome and Erythema multiforme)	new-onset bullous pemphigoid-like reaction delayed T-cell-mediated hypersensitivity reaction reactivation of HSV lichen planus-like reactions pernio erythema-like eruptions urticarial eruptions neutrophilic dermatosis leukocytoclastic vasculitis morbilliform eruptions delayed large local reactions
Girolami et al. [19]	Case report	1 F	mild interstizial mixed inflammatory infiltrate with neutrophils and rare eosinophils; extensive blood extravasation	livedoid skin reaction on proximal and distal lower limbs, abdomen, and proximal upper limbs
Dash et al. [20]	Case report	1 M	Intraepidermal infiltration of lymphocytes and neutrophils; degenerated/apoptotic keratinocytes; perivascular and periadnexal inflammatory cell infiltrate; extravasation of blood cells	multiple purpuric macules widespread; skin necrosis; oral erosions; haemorragic crusting (SJS)
Wang C.S. et al. [21]	Case report	1 M	foci of angulated parakeratosis, acanthosis, mild spongiosis, and mixed lympho-histiocytic and eosinophilic infiltrates	erythematous papules and plaques over the neck, trunk, back, and limbs
Fiorillo et al. [22]	Case report	1 F	Perivascular inflammatory infiltrate with intraparietal injury of vessels	purpuric macules and papules on both lower legs
Sandhu et al. [23]	Case series	2 (1 F-1 M)	1 case: purpuric extravasation of erythrocytes with damage of wall vessels with cellular debris 1 case: purpuric extravasation of erythrocytes with damage of wall vessels with cellular debris	1 case: purpuric rash on both lower legs palpable purpura distributed symmetrically over hands, forearms, gluteal region, and lower limbs
Bostan et al. [24]	Letter to editor	1 M	hyperkeratosis in the epidermis, perivascular mixed inflammation, and erythrocyte extravasation in the dermis	erythematous macules and palpable papules on the legs, forearms, and right belly
Vassallo et al. [25]	Letter to editor	1 F	lymphocytic vasculitis with plump endothelial cells	itchy maculopapular rash
Cohen et al. [26]	Letter to editor	1 F	perivascular mixed inflammatory infiltrate with numerous neutrophils, lymphocytes, and occasional eosinophils	palpable purpuric papules on the bilateral lower legs
Drago et al. [27]	Letter to editor	1 M	interface dermatitis with a lichenoid pattern	pruritic and burning eruptions on trunk and proximal extremities
Di Bartolomeo et al. [28]	Letter to editor	8 (6 F-2 M)	capillaritis without signs of vasculitis	symmetrical, slightly itchy, purpuric skin rash
Ambrogio et al. [29]	Case report	1 M	moderate lympho-monocytes infiltration with a perivascular distribution and with rare and focal involvement of the epidermis that presented mild spongiosis	purpuric spread on both legs/Majocchi’s disease
Rose et al. [30]	Case report	1 F	vacuolar interface alteration with superficial and mid-dermal perivascular and periadnexal lymphocytes	pruritic rash on injection site and arms
Gambichler et al. [31]	Systematic review with case series	1 M 1 (not specified) 1 (not specified) 1 M 1 F	subepidermal/intraepidermal bulla and eosinophils vacuolar interface dermatitis, including lymphocytic infiltrates along the dermo-epidermal junction associated with dyskeratoses of basal keratinocytes nuclear-dust and small vessels vasculitis purpuric extravasation of erythrocytes with damage of wall vessels with cellular debris perivascular lymphocytic infiltrates, mild vacuolar changes with occasional necrotic keratinocytes and erythrocyte extravasates	erythematous-bullous skin lesions Rowell’s syndrome (LE) leucocytoclastic vasculitis purpuric skin rash Majocchi’s disease-like rash

**Legend.** V-REPP: Vaccine-related eruption of papules and plaques. SJS: Steven-Johnson syndrome.

## Data Availability

Not applicable.

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
