# Peer review of "Histopathological Patterns of Cutaneous Adverse Reaction to Anti-SARS-CoV-2 Vaccines: The Integrative Role of Skin Biopsy"

_vaccines, 2023, doi:10.3390/vaccines11020397_

Round 1

Reviewer 1 Report

Adverse reactions due to Covid-19 vaccines   It is relevant, but not so interesting, because we had mant reports about it. This important is regarding in terms of histipathological evaluations and a review of all  reports   The paper is well written , but  abstract is chaotic( not easy to read),results are not clear to read,table should be added. should be shortened. Sentences are long and grammarily have faults in the abstract and in the results. Authors should revise.

1- Abstract includes too  long sentences to understand. Please make them more understandable

2-It would be better to know type of adverse reaction due to the type of vaccine.. A separate table including a list of adverse reactions according to vaccines may be fine.

3-Or  a table adverse reactions  with histopathological reactions may be arranged.(In the section of results)

4-Language should be revised

Author Response

Reviewer n'1: Adverse reactions due to Covid-19 vaccines   It is relevant, but not so interesting, because we had mant reports about it. This important is regarding in terms of histipathological evaluations and a review of all  reports   The paper is well written , but  abstract is chaotic( not easy to read),results are not clear to read,table should be added. should be shortened. Sentences are long and grammarily have faults in the abstract and in the results. Authors should revise.

1- Abstract includes too  long sentences to understand. Please make them more understandable.

Answer n'1: Dear Reviewer n'1, first of all thank you very much for this wonderful review. Yes, we proceeded to correct the abstract section with great attention to the lenght of the sentences. We hope it will be fine. Thanks a lot.

Reviewer n'1:

2-It would be better to know type of adverse reaction due to the type of vaccine.. A separate table including a list of adverse reactions according to vaccines may be fine.

3-Or  a table adverse reactions  with histopathological reactions may be arranged.(In the section of results)

Answer n'2: Dear Reviewer n'1, thanks again. So, we added a complete table (table 1) with all informations about the types of the cutaneous reactions and the histopathological patterns, with attention to the name of the author, number of patients and gender of patients.

Reviewer n'1:4-Language should be revised.

Answer n'3: Dear Reviewer, thank you. We had it corrected by a native english speaker. We hope that now manuscript is improved.

Reviewer 2 Report

Dear Authors

Manuscript is fine but need following corrections:

1. References need to be set as per author guidelines, there is difference in various references.

2. More articles need to be incorporated.

3. Some interesting facts needed to be included.

4. It's not very informative for the readers.

Author Response

Reviewer n'2

Dear Authors

Manuscript is fine but need following corrections:

1. References need to be set as per author guidelines, there is difference in various references.

2. More articles need to be incorporated.

3. Some interesting facts needed to be included.

4. It's not very informative for the readers.

Answer n'1: Dear Reviewer n'2, thank you very much for your important tips useful to improve the quality of our manuscript. We added some other papers that could be adaptable for our work and we have corrected the style of the references following MDPI guidelines. So, we hope that our manuscript, now, will be fine. Thanks again,

the authors

Reviewer 3 Report

The authors present a database search on histologically proven cutaneous reactions timely associated with the SARS-CoV2 vaccination. The manuscript is clearly written, but some points need to be addressed. In particular your data regarding histopathology and the limited number of cases compared to published data weaken the manuscript

1. What is the rationale behind only histologically proven diseases? The number of patients is very low due to the design of the study.

2. What is the normal incidence / prevalence of the dieseases, put them in relation to your findings.

3. Please discuss Gambichler et al., JEADV, 2022; Seirafianpour F et al., Dermatol. Therap., 2022 and many other groups.

4. Your title describes histological patterns, however you neither show a pattern, nor do you classify the biopsies. From by point of view the pattern is just due to the disease and not the vaccine.

Author Response

Reviewer n'3

The authors present a database search on histologically proven cutaneous reactions timely associated with the SARS-CoV2 vaccination. The manuscript is clearly written, but some points need to be addressed. In particular your data regarding histopathology and the limited number of cases compared to published data weaken the manuscript

  1. What is the rationale behind only histologically proven diseases? The number of patients is very low due to the design of the study.

Answer n'1: Dear Reviewer n'1, first of all, thank you very much for your useful and wonderful words to improve our manuscript. So, we decided to conduct a systematic review related only on cases with histopathological confirmation because, from a dermatopathological point of view, we wanted to highlight how much rarer papers describing histopathologic patterns beyond the usual clinical/morphologic features are to be found, initiating a reflection on what can be and whether histologic biopsy can really have only an "ACCESSORY" OR "DIAGNOSTIC" role in this setting. From this basic will, we purposely excluded the myriad scientific papers in which no histological biopsy had been performed. We realize that the number of patients was very limited in this way, but, that was precisely our purpose. We have added a small paragraph in the "limitations" section in which, to be fair, we state how considering only histologically confirmed cases reduced the sample size by no small amount. Thank you very much.

Reviewer n'3:

2. What is the normal incidence / prevalence of the dieseases, put them in relation to your findings.

3. Please discuss Gambichler et al., JEADV, 2022; Seirafianpour F et al., Dermatol. Therap., 2022 and many other groups.

Answer n'2: Dear Reviewer, thanks again.We fully understand his point of view and have moved in three different ways to respond to his valuable suggestions:
1.) We have read, studied, expanded and discussed in more depth the papers indicating and treating both skin reactions during Covid-19 and after administration of anti-SARS-CoV-2 vaccine. In addition, we dwelt on the prevalence of disease (as you indicated) proving and confirming that all reports included in this review had as their basis the temporal "link" between vaccine administration and subsequent (presumed) cutaneous ( and non) adverse reaction. In this way, in the discussion section, we have provided an explanation of this fundamental concept, incorporating the fact that the histopathological patterns described in the course of an adverse skin reaction are coadjuvant and descriptive of a manifestation that is first and foremost CLINICAL, but that can be decisive (or at any rate very important) both in the field of diagnostic confirmation and in the field of research 

Reviewer 4 Report

The article states that is a meta-analysis of reports regarding dermatological adverse events; however, the authors just reported, in a table, a summary of the finding reported. How many patients were screened in general, and how many individuals had a dermatological adverse effect? How many of those reported an autoimmune disease after vaccination?  Which vaccine was responsible for the highest frequency of adverese reactions? What are the limitations of the study? The limitations of the study should be included. 

Author Response

Dearest Reviewer #4, thank you very much, first of all, for your valuable advice to improve the quality of our manuscript. We have addressed all your critical issues in the following points:
1) We added the number of patients included in the review, with attention to the number of patients who had developed an autoimmune-like manifestation/pathology, which we promptly reported in the major text;
2) We added information regarding the types of vaccines most commonly used in the papers we selected according to our inclusion criteria and, when necessary, discussed their possible mechanisms of etiopathogenesis.
3) We added a new paragraph with the limitations found in our paper.

Round 2

Reviewer 2 Report

Corrections done, The paper could be accepted in present form

Reviewer 3 Report

Most points raised by the reviewers were adressed.

Reviewer 4 Report

The authors modified the manuscript as requested. Now the manuscript is suitable for publication.